# Evaluation of the Immunomodulatory Effect of the Recombinant 14-3-3 and Major Antigen Proteins of *Strongyloides stercoralis* against an Infection by *S. venezuelensis*

**DOI:** 10.3390/vaccines10081292

**Published:** 2022-08-10

**Authors:** Liz F. Sánchez-Palencia, María Trelis, Julio López-Abán, Alicia Galiano, Belén Vicente, Esther del Olmo, Antonio Muro, Dolores Bernal, Antonio Marcilla

**Affiliations:** 1Àrea de Parasitologia, Departament Farmàcia i Tecnologia Farmacèutica i Parasitologia, Universitat de València, 46100 Burjassot, Spain; 2Unidad Mixta de Endocrinología, Nutrición y Dietética Clínica, Instituto de Investigación Sanitaria La Fe, 46026 Valencia, Spain; 3Grupo de Enfermedades Infecciosas y Tropicales (e-INTRO), IBSAL-CIETUS (Biomedical Research Institute of Salamanca-Research Centre for Tropical Diseases at the University of Salamanca), Facultad de Farmacia, Universidad de Salamanca, Méndez Nieto s/n, 37007 Salamanca, Spain; 4Departamento de Ciencias Farmacéuticas: Química Farmacéutica, Facultad de Farmacia, Universidad de Salamanca, IBSAL CIETUS, 37007 Salamanca, Spain; 5Departament de Bioquímica i Biologia Molecular, Universitat de València, 46100 Burjassot, Spain

**Keywords:** *Strongyloides stercoralis*, 14-3-3, major antigen, vaccination, ADAD, cytokines

## Abstract

Strongyloidiasis, caused by *Strongyloides stercoralis,* is a neglected parasitic disease that represents a serious public health problem. In immunocompromised patients, this parasitosis can result in hyperinfection or disseminated disease with high levels of mortality. In previous studies, the mRNAs encoding for the 14-3-3 and major antigen proteins were found to be expressed at high levels in *S. stercoralis* L3 larvae, suggesting potential key roles in parasite-host interactions. We have produced them as recombinant proteins (rSs14-3-3 and rSsMA) in a bacterial protein expression system. The serum levels of anti-rSs14-3-3 and anti-rSsMA IgGs are increased upon infection with *S. venezuelensis,* validating the use of the mouse model since the native 14-3-3 and MA proteins induce an immune response. Each recombinant protein was formulated in the adjuvant adaptation (ADAD) vaccination system and injected twice, subcutaneously, in CD1 mice that were experimentally infected with 3000 *S. venezuelensis* L3 to evaluate their protective and immunomodulatory activity. Our results, including the number of parthenogenetic females, number of eggs in stool samples and the analysis of the splenic and intestinal indexes, show that the vaccines did not protect against infection. The immunization with rSs14-3-3 induced changes in the cytokine profile in mice, producing higher expression of IL-10, TGF-β, IL-13 and TNF-α in the spleen, suggesting a Th2/Treg-type response with an increase in TNF-α levels, confirming its role as an immunomodulator.

## 1. Introduction

Human strongyloidiasis is an infection caused by *Strongyloides stercoralis* (and rarely *S. fülleborni*) which, like other soil-transmitted helminthiases (STHs), is acquired through direct contact with contaminated soil [1]. Some of these helminths are considered cosmopolitan although, according to epidemiological data, their location is mostly limited to tropical and subtropical areas, affecting the poorest and most disadvantaged communities [2]. The World Health Organization (WHO) [1] estimates that between 30 and 100 million individuals are infected worldwide by strongyloidiasis; however, these data may be underestimated because this disease requires special laboratory methods for diagnosis. Moreover, drugs other than those recommended by the WHO are routinely used for the treatment of the STHs, which are not included in control programs [3]. This disease is caused by direct penetration of the *S. stercoralis* infective larvae through human skin in contact with contaminated soil during agricultural, domestic, and recreational activities. Walking barefoot is therefore a major risk factor for acquiring the infection [1]. The L1 rhabditiform larvae eliminated with the feces are transformed in the soil into the L3 filariform larvae with infective capacity able to penetrate through the skin and initiate the parasitic cycle. The life cycle of *S. stercoralis* is unique and complex, combining two possibilities, free life cycles (with males and females outside the host) and parasitic cycles within the host (with intestinal parthenogenetic females) [4]. Its capacity for self-infection and multiplication in the host is unique, especially in immunocompromised patients in which strongyloidiasis derives in hyperinfection or in a disseminated disease with high levels of mortality [5]. The genus *Strongyloides* contains around 60 species. Two of these, *S. ratti* and *S. venezuelensis*, can experimentally infect laboratory rodents [6]. In fact, *S. venezuelensis* exhibits a life cycle in rats and mice that is similar to that observed for *S. stercoralis* in humans. For this reason, it is often used as a model for human strongyloidiasis, particularly for immunological studies [7].

Autochthonous strongyloidiasis has been detected in Spain since 1895, especially in Mediterranean regions, such as Valencia and Murcia. Currently only residual cases remain. Factors associated with autochthonous cases are agricultural activities mainly associated with the growing of rice and cleaning of irrigation ditches, or advanced age [8,9,10,11,12]. Moreover, the number of imported cases, mainly from Cambodia and Latin America, are increasing due to the change in the migratory flow [8]. Imported strongyloidiasis is a current and universal challenge as an imported disease in developed healthcare systems.

The control of strongyloidiasis in endemic areas with deficient healthcare systems continues to require efforts in prevention and treatment. The availability of a vaccine would be of great interest since it would limit the intensity of the disease burden and control transmission. With the aim of identifying potential new markers that could be used in disease diagnosis, vaccination, and treatment, we previously reported a proteomic analysis of the third larval stage of *S. stercoralis* obtained from patients in Valencian hospitals [13]. In the transcriptome study of this stage, the mRNAs encoding the 14-3-3 and major antigen proteins were present at high levels, suggesting that these proteins might be involved in the entry of the parasite through the skin of the host and participate in modulating the immune response [14]. Little is known about the role of the 14-3-3 protein family in *S. stercoralis* pathogenicity. Most researchers have suggested that 14-3-3 proteins may transmit regulatory signals related to cell physiology, such as proliferation, migration, and morphological changes during the parasite life cycle [15]. These proteins are considered to be good targets for anthelmintic drugs and vaccines, as well as candidates to develop new diagnostic methods [16,17].

To explore and evaluate the immunomodulatory activity and the vaccine and the diagnostic potential of *S. stercoralis* 14-3-3 (Ss14-3-3) and major antigen (SsMA) proteins in an experimental infection with *S. venezuelensis* in female CD1 mice, we produced them as recombinant proteins (rSs14-3-3 and rSsMA) in bacteria and formulated them in an adjuvant adaptation (ADAD) vaccination system. This vaccination system has been successfully used with similar antigens, for example a 14-3-3 protein from *Schistosoma bovis* and FABP from *Fasciola hepatica* [18,19].

## 2. Materials and Methods

### 2.1. Ethics Statements

This study was performed in strict accordance with the protocols approved by the Ministry of Agriculture and Livestock of the Government of the Community of Castilla y León, Spain (resolution n° 335 of 20 February 2019). The basic rules, applicable to the protection of the animals used in the experimentation that took place in the Animal Research and Welfare Service of the University of Salamanca, were followed. All surgeries were performed under pentobarbital anesthesia and every effort was made to minimize suffering.

### 2.2. S. venezuelensis and Experimental Mice Model

*Strongyloides venezuelensis* was obtained from a strain originally used in the Department of Parasitology, University of Minas Gerais, Belo Horizonte, Brazil. This strain was maintained by serial passages in Wistar rats routinely infected in the laboratory of Parasitic and Molecular Immunology, CIETUS, University of Salamanca. Rats were subcutaneously infected with third-stage larvae (L3) of *S. venezuelensis*, as previously described [20]. CD1 female experimental mice and maintenance rats were housed and maintained at the Research and Animal Welfare Service of the University of Salamanca under standard conditions, at a temperature of 24 °C and a humidity-controlled environment with 12 h day-night cycles and provided with water and food ad libitum.

### 2.3. Cloning of S. stercoralis Recombinant 14-3-3 and Major Antigen Proteins

The purification of *S. stercoralis* L3 larvae RNA has been previously described [14]. Briefly, total RNA was obtained with a Real Total RNA Spin Plus kit (Real Life-Sciences Solutions, Valencia, Spain), following the manufacturer’s protocol. The integrity of the RNA was verified by agarose gel (1%) electrophoresis. The RNA template was then used for cDNA synthesis with a High-Capacity cDNA Reverse Transcription Kit (Applied Biosystems, Waltham, MA, USA), and performed in a C1000 thermal cycler (Bio-Rad, Hercules, CA, USA). The cDNAs encoding Ss14-3-3 and SsMA were amplified by PCR with specific primers designed from the known nucleotide sequence of the parasite (GenBank codes CEK45735 and CEK45729, respectively). The Ss14-3-3 primers were: F: 5′-ACGGATCCATGGCTGAAAATAAGGATGAAC-3′ and R: 5′-AGCTGCAGCCATTGGTATCAGATGTC-3′; and F: 5′-ACGGATCCATCTGATGCAGCTAT-3′ and R: 5′-AGCTGCAGTAGCATTGTCATCGTG-3′ for the SsMA. The DNA encoding Ss14-3-3 and SsMA were successfully amplified and subcloned into the bacterial expression vector pQE30-32, containing a His-tag at the C-terminus (QIAGEN, Hilden, Germany). Transformed bacteria were selected by growth on LB agar medium with ampicillin. Plasmid DNA was isolated from candidate clones and the constructs were analyzed by sequencing using SeqMan Pro and SeqBuilder (Lasergene, DNASTAR Inc., v.7.0.0, Madison, WI, USA). The correct open reading frames of the sequences were confirmed using the ExPASy bioinformatics tool (Expert Protein Analysis System, http://www.expasy.org; accessed on 1 February 2021) [21], and homology with 14-3-3 (LM524971.1) and major antigen (LM525011.1) of *S. venezuelensis* was confirmed by comparison with the sequences reported in the NCBI (National Center for Biotechnology Information; www.ncbi.nih.gov, accessed on 1 February 2021) database, revealing a 82.38% identity for MA proteins and a 97.46% identity for the 14-3-3 homologues of *S. venezuelensis* and *S. stercoralis*.

### 2.4. Expression and Purification of rSs14-3-3 and rSsMA Proteins in E. coli

Individual clones of transformed *Escherichia coli* M15 cells were grown in LB medium with ampicillin until they reached an optical density at 600 nm of 0.6, and then the recombinant proteins were induced by the addition of isopropyl-β-D-1-thiogalactopyranoside to a final concentration of 1 mM (IPTG, Promega, Madison, WI, USA) for 3 h. The recombinant proteins were then purified by Ni-NTA Spin Column System (Qiagen, Hilden, Germany). The protein concentrations were determined using a Bradford assay (Bio-Rad, Hercules, CA, USA). Recombinant protein samples were analyzed by SDS-PAGE electrophoresis using nUView Precast Gels Tris-Glycine NB 4–20% (NuSep Inc., Germantown, MD, USA) and stained with Coomassie blue. Precision Plus Protein^TM^ Dual Color Standards (BIO-RAD, Hercules, CA, USA) were used as the molecular weight marker.

### 2.5. Vaccination Experiment Protocol

A total of 35 female CD1 mice, 8 weeks of age, were used for the immunization experiments. Mice were randomly divided into five experimental groups: control (neither immunized nor infected; *n* = 3), infection control (infected; *n* = 8), ADAD control (injected only with ADAD and infected; *n* = 8), rSs14-3-3 group (immunized with rSs14-3-3 in ADAD and infected; *n* = 8); and rSsMA group (immunized with rSsMA in ADAD and infected; *n* = 8). Animals were immunized subcutaneously with 10 µg of rSs14-3-3 or 10 µg of rSsMA emulsified in the ADAD vaccination system (a combination of the non-hemolytic saponins from *Quillaja saponaria* and a synthetic immunomodulator diamine AA0029 with the non-mineral oil Montanide ISA 763AVG) to obtain a long-term delivery [22]. Immunization was performed twice at an interval of two weeks. Seventeen weeks after the immunization, animals were infected with 3000 *S. venezuelensis* L3 larvae subcutaneously. ADAD is a long-term delivery adjuvant system with anti-inflammatory properties. Therefore, the vaccination challenge interval was increased to avoid the interference from the adjuvant described previously [23]. In addition, a wide vaccination-challenge interval is necessary for a vaccine, since in an ideal case, immunity should remain for a long time. Stool samples were obtained on the 5th, 6th, and 7th day post-infection. Blood samples were taken before and at the end of the assay (one-week post-infection). Serum was stored at −20 °C for indirect ELISA. After sacrificing the mice, spleens and intestines were collected and weighed to calculate organ-to-body weight ratios. The splenic index (SI) was calculated using the formula: SI = spleen weight (g)/body weight (g) × 100 [24], and the intestinal index (II) was calculated in a similar way. Parthenogenetic females were recovered from the first third of the small intestine. Spleens were stored in RNAlater (Sigma-Aldrich, St. Louis, MO, USA) at −80 °C for further gene expression analysis.

### 2.6. Detection of Antibodies in Sera by Indirect ELISA

Serum levels of anti-14-3-3 and anti-MA IgG antibodies were measured individually for each mouse in duplicate, by indirect ELISA, using the recombinant protein as a coating antigen. Briefly, 96-well polystyrene microtiter plates were coated overnight at 4 °C with 2 µg/mL of protein in carbonate-bicarbonate buffer (pH 9.6). The coated plates were washed three times with PBS with 0.05% Tween 20 (PBST) and uncoated sites were blocked with 5% fat-free dry milk in PBST for 1 h at 37 °C. 100 µL of sera diluted 1/100 in PBST were added to each well and incubated at 37 °C for one hour. The plates were washed three times with PBST, and then incubated with 100 µL HRP-conjugated antibody goat anti-mouse IgG (1/2000) (Sigma-Aldrich, St. Louis, MO, USA) as the secondary antibody. The enzymatic color reaction was generated using a solution with chromogen orthophenylenediamine 0.04% in phosphate-citrate buffer (0.2 M Na_2_HPO_4_, 0.1 M citric acid; pH 5) and 0.001% H_2_O_2_ as a substrate. The absorbance was measured at 490 nm using an iMark™ Microplate Absorbance Reader (Bio-Rad, Hercules, CA, USA).

### 2.7. RNA Isolation, Reverse Transcription and Quantification of Cytokine mRNAs

Total RNA from spleens was purified using a Total RNA Spin Plus Purification kit (Real Life-Sciences Solutions, Valencia, Spain), following the specific protocol for tissues provided by the manufacturer. The quantity and quality of the RNA were determined by absorbance at 260/280 nm. The cDNA was synthesized using the High-Capacity cDNA Reverse Transcription kit (Applied Biosystems, Waltham, MA, USA), according to the manufacturer’s protocol. The cytokine mRNAs (IFN-γ, TNF-α, IL-12, IL-4, IL-5, IL-13, IL-17, IL-10, TGF-β, IL-23) were analyzed by quantitative PCR (qPCR) using TaqMan^TM^ Gene Expression Assays containing a specific pair of primers and 6-FAM™ dye-labeled probe (Applied Biosystems, Waltham, MA, USA). For quantitative PCR, 40 ng of cDNA were added to 10 µL of TaqMan^TM^ Fast Advanced Master Mix (Applied Biosystems, Waltham, MA, USA), 1 µL of the specific gene expression assay and water to a final reaction volume of 20 µL. To perform the amplification reactions in 96-well plates, the StepOne™ Real-Time PCR System (Applied Biosystems, Waltham, MA, USA) was employed, using the following thermal cycler conditions: 2 min for activation at 50 °C followed by 10 min at 95 °C, and 40 cycles of 15 s denaturation at 95 °C and finally 1 min of anneal/extend at 60 °C. In each plate, an endogenous and a negative control were analyzed in duplicate. The quantification was performed using β-actin as the housekeeping gene.

The cycle threshold (Ct) value was calculated for each sample. To estimate the influence of the immunization and/or infection in the expression levels of cytokines, a comparative quantitation method (2^−∆∆Ct^—method) was used [25].

### 2.8. Statistical Analysis

All statistical analyses were performed using a GraphPad Prism v9 for Windows (GraphPad Software, San Diego, CA, USA, www.graphpad.com, accessed on 1 February 2021). Results were expressed as the mean ± the standard error of the mean (SEM). One-way and two-way analyses of variance (ANOVA), followed by post-hoc Tukey’s multiple comparisons test, were used to compare significant differences between different conditions. The tests used are indicated in each figure legend. Statistical significance was established at *p* < 0.05.

## 3. Results

### 3.1. Cloning, Expression and Purification of Recombinant Ss14-3-3 and SsMA Proteins from S. stercoralis

The DNA encoding Ss14-3-3 (GenBank n° LN715176.1) and SsMA (GenBank n° LN715173.1) was successfully amplified and subcloned into the expression vector. The recombinant His-tagged proteins were purified under denaturing conditions, obtaining a final concentration of 1 mg/mL for rSs14-3-3 and 0.77 mg/mL for rSsMA. SDS-PAGE analysis showed that both displayed the expected molecular weight (26.5 KDa and 26.7 KDa, respectively) (Figure 1).

### 3.2. Immunization with rSs14-3-3 and rSsMA Proteins Do Not Induce Effective Protection against the Experimental Infection

To study the potential of the rSs14-3-3 and rSsMA proteins to protect mice against infection with *S. venezuelensis*, we first determined the number of eggs per gram (EPG) in feces (Figure 2), as an indicator of worm burden in the small intestine. Eggs were observed in feces after 5 days of infection. The comparison of the number of EPG between groups on the same day post-infection revealed an increase in immunized groups compared to the infection control group, on the seventh day post-infection for the rSs14-3-3 (387,506.439 ± 49,571.363 vs. 350,851.027 ± 82,665.61), and on the sixth day post-infection for rSsMA (350,851.027 ± 82,665.61 vs. 161,666.79 ± 27,739.972) (Figure 2).

When the number of recovered parthenogenetic females was determined and compared between groups, an increase in the number of females was observed in both groups immunized with the recombinant proteins (rSs14-3-3: 222.418 ± 30.616 and rSsMA: 243.016 ± 35.251) (Figure 3), which was consistent with the previous results for the EPG.

### 3.3. Splenic and Intestinal Indexes Reflect the Non-Protection of the Vaccine against Infection

The splenic index indicates an immunological cellular response activation after infection or vaccination. The spleen volume decreases in weight and size with age, but increases upon infection [26]. When analyzing the effect of immunization in the splenic index, the administration of the rSs14-3-3 and rSsMA proteins produce a slight increase in this index with respect to the infection and ADAD controls (Figure 4A).

The intestinal index indicates the local inflammatory effects of the infection by *S. venezuelensis*. The intestinal indexes present a similar value and are slightly increased in both mice groups immunized with the recombinant proteins when compared to the infection and ADAD controls (Figure 4B). The differences between experimental groups are not statistically significant, suggesting that the recombinant proteins did not prevent local inflammation.

### 3.4. Humoral Antibody Response to the Immunization of Mice with rSs14-3-3 and rSsMA Proteins and/or Infection with S. venezuelensis

To investigate the capacity of rSs14-3-3 and rSsMA proteins to induce an antibody response, specific IgG levels in serum were evaluated by indirect ELISA before and after the treatment (immunization and infection) (Figure 5). All mice presented higher specific IgG antibody levels against the corresponding protein after the immunization with rSs14-3-3 or rSsMA and/or after the infection. The immunization did not increase the IgG levels after 7 days of infection.

In addition, we tested the generation of antibodies against *S. venezuelensis* 14-3-3 and major antigen proteins in the mouse model and could show that they increase upon infection (Figure 5). This result further validates the use of the mouse model to simulate relevant responses comparable to the human infection and suggests that the genes encoding these proteins are likely to be induced upon *S. venezuelensis* infection. Thus, both the level of similarity between the homologues and the observation that infection induces the production of antibodies specific for these two proteins validate the experimental design chosen here.

### 3.5. Cytokine Profile in Mice after Immunization with rSs14-3-3 and rSsMA and Experimental Infection

To further investigate the type of response that was induced by immunization with rSs14-3-3 and rSsMA, followed by the infection, the levels of expression of selected cytokines (IFN-γ, TNF-α, IL-12, IL-4, IL-5, IL-13, IL-17, IL-10, TGF-β, IL-23) were measured in the spleen using qPCR. Higher expression levels of IL-10, TGF-β, IL-13 and TNF-α were detected in the rSs14-3-3 group when compared to control groups, which confirmed its immunomodulatory capacity (Figure 6). In the rSsMA group, there are no differences in the expression of the cytokines analyzed; however, the expression of IL-10 and IL-13 presents a slight increase compared to controls, but the results were not statistically significant (Figure 6).

## 4. Discussion

Strongyloidiasis is a neglected tropical disease with no available vaccines, and it is not included in WHO control programs. There is a need for vaccine development due to the potential ability of *S. stercoralis* to replicate within the host as part of their autoinfection cycle, which leads to persistent infections. These situations can cause uncontrolled multiplication and dissemination, where larvae spread to and affect all the internal organs of people with compromised immune systems [27,28]. Identification and characterization of *S. stercoralis* immunogenic proteins are usually carried out by proteomic analysis on crude helminth homogenates, which are then chosen based on immunogenicity, ability to stimulate an immune response, and minimization of potential secondary effects [2,4,29]. Recent studies have predicted promising candidates in proteomic analyses of *S. stercoralis* using bioinformatic tools [13,17,30]. Our group was pioneering in detailing the transcriptome of the infective *S. stercoralis* L3 larvae [14]. In that study, two mRNAs, corresponding to major antigen and 14-3-3 proteins, were the most abundant transcripts found in this larval phase [14]. In the present study, we produced and purified these two proteins in bacteria and evaluated their potential protective and immunomodulatory capacity. We used the adjuvant adaptation (ADAD) vaccination system, developed and tested for vaccination against other helminth infections, such as fasciolosis and schistosomiasis, to improve the limitations of classical Freund’s adjuvant [22,31,32].

Our results indicate that the assayed recombinant proteins did not produce an efficient protection against the experimental infection using the *S. venezuelensis* mouse model, despite the homology between proteins 14-3-3 and the major antigen of both species. Surprisingly, the numbers of parthenogenetic females recovered from the small intestine and the numbers of EPG were higher in immunized animals, as compared to the control groups. These results suggest that not only did the immunization not protect from the infection, but that the infection was favored by the immunization. The presence of eggs in feces is related to the successful establishment and development of parasites. In mice immunized with exosomes from *Echinostoma caproni*, the number of eggs in feces was reduced after 6 weeks of infection. These data suggest that the immunization produced a delay in the development of the parasite, ameliorating the symptoms of the infection, but not preventing the infection [33]. In a different immunization study with Sj-p80 in Montanide^TM^ ISA61VG adjuvant carried out in mice infected with *Schistosoma japonicum*, there was a reduction in the female burden, but the number of eggs did not decrease. However, the viability/hatching rates of the eggs were reduced [34]. A vaccination system using the 14-3-3 protein from *Schistosoma bovis* and FABP from *Fasciola hepatica* yielded high protection in terms of parasite burden and liver damage [19,22].

The splenic index increases upon infection. In this sense, Mei and colleagues found elevated splenic indexes in mice infected with *Plasmodium bergei* and this index was higher in mice co-infected with *Trichinella spiralis* [24]. In addition, immunosuppressed mice infected with *Sporothrix schenckii*, an opportunistic fungus, presented higher splenic indexes [26]. In our study, the infected groups previously immunized with the recombinant proteins showed higher indexes, suggesting a stimulation of this organ. The intestinal index indicates inflammation associated with the parasitic infection. The presence of a higher number of parthenogenetic females in immunized groups explains the increase observed in the intestinal index.

Recombinant proteins did not induce any protection against helminth infection; however, the immunization with the rSs14-3-3 previous to infection elicited a higher expression of the anti-inflammatory cytokines IL-13, IL-10 and TGF-β associated with a Th2/Treg response in the spleen, which could be involved in promoting the establishment of the parasite and the success of the infection. IL-13 is considered to be an important cytokine in the immunoregulation of inflammatory, allergic and autoimmune processes [35,36,37]. Control of *Strongyloides* spp. infections generally requires type 2 immune responses that have the advantage of causing reduced collateral damage, as compared to Th1 or Th17 responses. Type 2 cytokine responses (particularly IL-13) play an important role in fibrosis and wound healing and they are postulated to help healing tissue damage induced by many tissue-invasive helminth parasites [36]. In addition, in response to IL-4 and IL-13, macrophages differentiate to M2 macrophages, a type of alternatively activated macrophage that is also anti-inflammatory, producing IL-10 and TGF-β [37]. IL-13, together with IL-10, can inhibit the production of pro-inflammatory cytokines in macrophages [35]. In addition, helminth infections are also intricately associated with the increased expression of the regulatory cytokines IL-10 and TGF-β [38]. The overexpressed cytokines IL-10 and TGF-β could lead to immunological tolerance and certain types of damage in the host, but at the same time, they could prevent the expulsion of worms, thereby promoting chronic helminth infections [37]. The mechanisms by which parasite-secreted molecules restrain host immune responses have been investigated in various mouse models. A requirement common to all progressing parasitic infections is that parasites must be able to evade the full effects of host immune responses and survive in the host for long periods [38,39]. Further experiments with NK IL10 and TGF-β knock-out mice will help to clarify the responsibility of the overexpressed cytokines in the effect of the evolution of the pathology of the parasite.

*S. stercoralis* is a parasite able to replicate in the human host, carrying out autoinfection cycles. Although autoinfection is limited by the host immune response, potentially fatal hyperinfection with a disseminated disease can occur when immunity is compromised [4,40]. The immunization with the rSs14-3-3 protein has also elicited, in the spleen, a higher expression of the TNF-α, which is considered to act as a pleiotropic cytokine, in addition to being a pro-inflammatory cytokine. TNF-α has a critical role in the regulation of Th2 cytokine-mediated host protection against helminth infections [41]. Some cases of hyperinfection by *S. stercoralis* have been reported in patients in immunosuppressed states, due to the administration of anti-TNF-α to treat autoimmune diseases. Therefore, high levels of TNF-α could prevent the hyperinfection syndrome caused by *S. stercoralis*, which is associated with high mortality [40,42,43]. Accordingly, patients should be tested for strongyloidiasis prior to the start of an immunosuppressive treatment to identify asymptomatic carriers from endemic regions.

An interesting result was obtained when analyzing specific IgG antibodies against rSs 14-3-3 and rSsMA in sera from mice infected with *S. venezuelensis.* Even though the immunization did not potentiate the humoral immune response produced by the infection, our results suggest that the infection itself is inducing a humoral response against these two native proteins. This confirms that 14-3-3 and MA proteins are among the more abundant and immunogenic proteins of the parasite, as described previously in the transcriptomic analysis of the third larval stage of *S. stercoralis* [14]. This observation further validates the use of the mouse model to simulate relevant responses comparable to the human infection and suggests that the genes encoding these proteins are likely to be induced upon *S. venezuelensis* infection. Thus, both the level of similarity between the homologues and the observation that infection induces the production of antibodies specific for these two proteins validate the experimental design chosen here. Until now, few recombinant proteins have been studied as potential immunogenic molecules for serodiagnosis of strongyloidiasis [16,44]. The antibodies against rSs14-3-3 and rSsMA are not increased with the immunization and they then do not provide any protective effect, but their quantification could be useful for diagnosis and as a marker of the severity of the infection [44,45].

We know that one of the limitations of the study is the lack of an accessible experimental model for *S. stercoralis*, but the high homology between *Strongyloides* species in the selected proteins (14-3-3 and MA) allowed us to use the *S. venezuelensis* model in mice demonstrating that the immunization with rSs14-3-3 and rSsMA in ADAD system can modulate the immune response of the host, favoring *S. venezuelensis* infection instead of protecting it; therefore, they cannot be used as potential vaccine candidates. However, both, rSs14-3-3 and rSsMA can be used in ELISA assays to diagnose strongyloidiasis, since they have detected specific IgGs in the serum of infected mice. Further experiments to investigate the immune response at longer times after the immunization and infection are needed to assess the production of other specific cytokines and antibody isotypes to confirm the suggested immunomodulation.

## 5. Conclusions

Immunization with recombinant Ss14-3-3 and SsMA did not prevent the *S. venezuelensis* infection, but they do promote immunomodulatory effects that appear to favor the establishment and development of parasite females. The recombinant proteins have the potential to be used in ELISA assays to diagnose the disease since the serum levels of anti-Ss14-3-3 and anti-SsMA IgGs both increased in strongyloidiasis. In addition, immunization with the rSs14-3-3 protein activates the immune response with an induction of a Th2/Treg-type response together with a high production of TNF-α, confirming its role as an immunomodulator.

## Figures and Tables

**Figure 1 vaccines-10-01292-f001:**
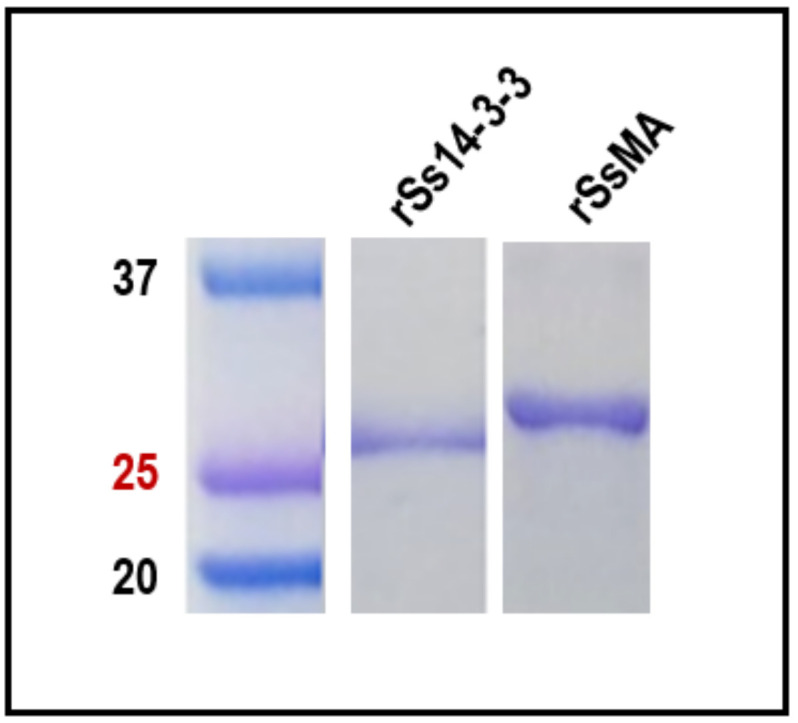
Synthesis and purification of rSs14-3-3 and rSsMA in *E. coli.* Purified recombinant Ss14-3-3 and SsMA proteins (10 µg/lane) were separated on SDS-PAGE and visualized using Coomassie blue stain. Molecular weight markers in kDa are shown (Precision Plus Protein^TM^ Dual Color Standards (BIO-RAD, Hercules, CA, USA)).

**Figure 2 vaccines-10-01292-f002:**
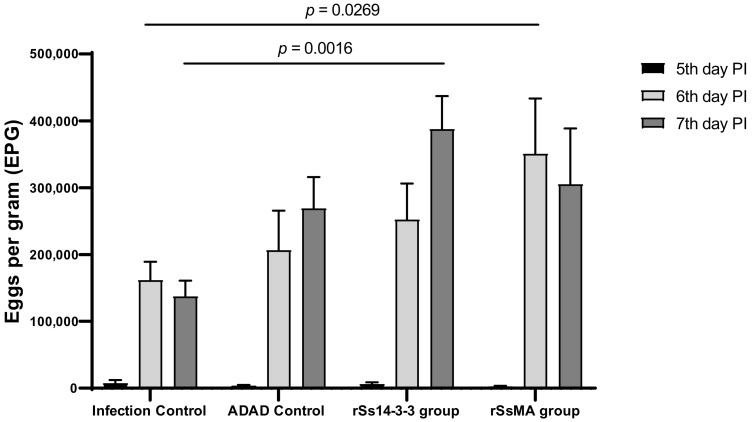
Number of eggs per gram (EPG) in mice feces after immunization with rSs14-3-3 and rSsMA formulated in ADAD vaccination system and challenged with 3000 *S. venezuelensis* L3 determined at 5, 6, and 7 days post-infection (PI). Bars represent means ± SEM. Differences between groups were determined by two-way ANOVA F_(3,84)_ = 4.18; *p* = 0.0083. Post-hoc Tukey’s honestly significant difference (HSD) test *p*-values are shown in the graph.

**Figure 3 vaccines-10-01292-f003:**
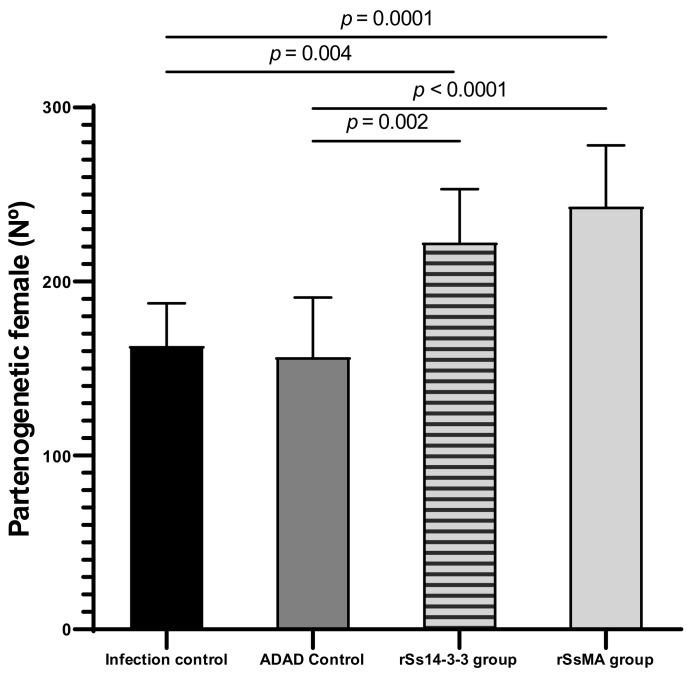
Number of *S. venezuelensis* parthenogenetic females after immunization with rSs14-3-3 and rSsMA formulated in ADAD vaccination system and challenged with 3000 L3. Bars represent means ± SEM. The number of recovered females under each experimental condition was evaluated by one-way ANOVA analysis with respect to the infection control group (F_(3,27)_ = 1; *p* < 0.0001). Post-hoc Tukey’s honestly significant difference (HSD) test *p*-values are shown in the graph.

**Figure 4 vaccines-10-01292-f004:**
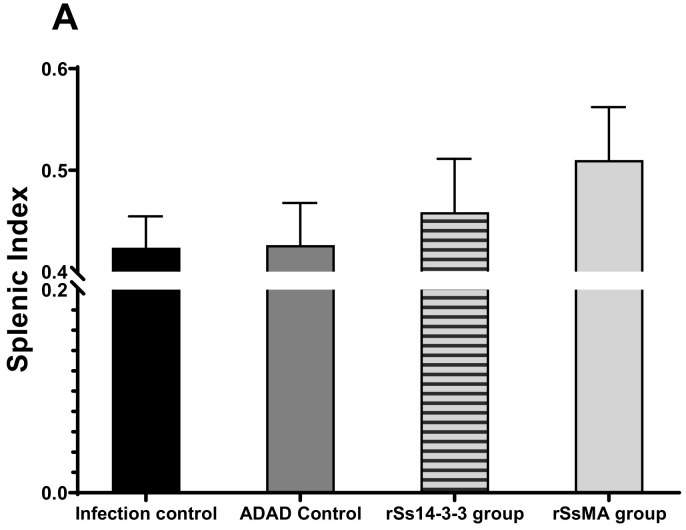
Effect of the immunization in the Splenic (**A**) and Intestinal (**B**) indexes after infection of mice with *S. venezuelensis*. Bars represent means ± SEM. Data were evaluated by one-way ANOVA analysis against infection and ADAD controls, but there were not significant differences (F_(3,28)_ = 0; *p* = 0.5086 for the splenic index and F_(3,28)_ = 0.5649; *p* = 0.6427 for the intestinal index).

**Figure 5 vaccines-10-01292-f005:**
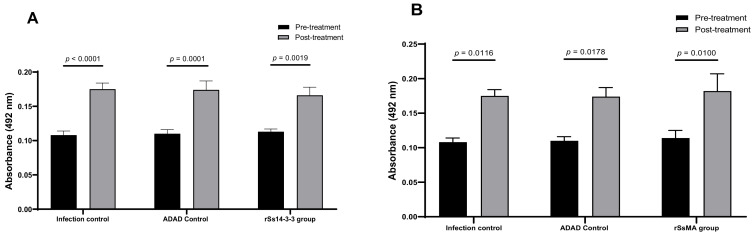
Serum levels of anti-14-3-3 IgG (**A**) and anti-MA IgG (**B**) in mice before and after the treatment (immunization and infection). IgG levels were measured by indirect ELISA by coating the plates with 2 µg/mL of 14-3-3 protein or 2 µg/mL MA protein per well, respectively. (**A**) IgG levels against rSs14-3-3 protein; (**B**) IgG levels against rSsMA protein. Bars represent means ± SEM. Differences between groups were determined by two-way ANOVA. The IgG levels are significantly increased after the assay ((**A**): F_(1,42)_ = 70.24; *p* < 0.0001; (**B**): F_(1,42)_ = 37.08; *p* < 0.0001). Post-hoc Tukey’s honestly significant difference (HSD) test *p*-values are shown in the graph.

**Figure 6 vaccines-10-01292-f006:**
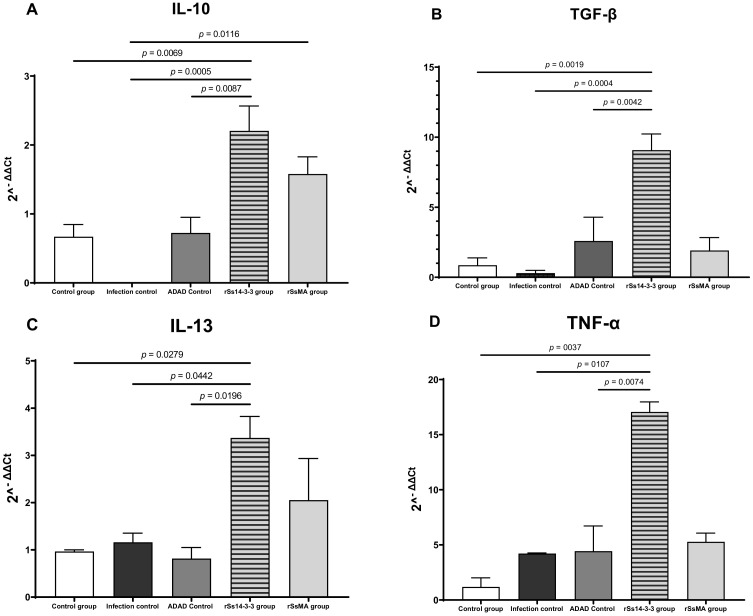
Expression levels of IL-10, TGF-β, IL-13 and TNF-α in spleen of mice immunized with rSs14-3-3 and rSsMA formulated in ADAD and challenged with 3000 *S. venezuelensis* infective L3. (**A**) IL-10 mRNA levels; (**B**) TGFβ mRNA levels; (**C**) IL-13 mRNA levels; (**D**) TNFα mRNA levels. Bars represent means ± SEM. Data were evaluated by one-way ANOVA analysis for each cytokine IL-10 (F_(4,9)_ = 13; *p* = 0.0007); TGF-β (F_(4,19)_ = 1; *p* = 0.0001); IL-13 (F_(4,10)_ = 5; *p* = 0.0152; TNF-α (F_(4,6)_ = 12; *p* = 0.0042). Post-hoc Tukey’s honestly significant difference (HSD) test *p*-values are shown in the graphs.

## Data Availability

Not applicable.

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
