# Peer review of "Evaluation of the Immunomodulatory Effect of the Recombinant 14-3-3 and Major Antigen Proteins of Strongyloides stercoralis against an Infection by S. venezuelensis"

_vaccines, 2022, doi:10.3390/vaccines10081292_

Round 1

Reviewer 1 Report

Reviewer’s comments

This study the protective potential and immunomodulatory capacity of recombinant protein 14-3-3 and major antigen protein of Strongyloides stercoralis against the infection by S. veneauelensis. The authors reported that the two recombinant proteins couldn’t be able to induce protective immunity, but could increase the serum levels of anti-rSs14-3-3 and anti-rSsMA IgGs and a higher expression of IL-10, TGF-β, IL-13 and TNF-α in spleen of immunized/infected mice. The authors concluded that these two proteins have the potential to be used for diagnosis and in preventing hyperinfection processes. In my opinion, the experimental design of this study has some big flaws and the results haven’t been clearly described.

1.       In this study, two proteins of S. stercoralis were used, but not against the infection of the same species, other than a different species. However, the information on the sequence identities between Ss-14-3-3 and Sv-14-3-3 as well as between Ss-MA and Sv-MA wasn’t provided. In addition, although the transcription of Ss-14-3-3 and Ss-MA encoding genes is abundant in iL3 of S. stercoralis, it is not known whether this is the same case in S. veneuelensis. Therefore, the reason for selecting these two molecules was not sufficient.

2.       There is a lack of information on the changes of anti-rSs14-3-3 and anti-rSsMA IgG levels and the transcription levels of selected cytokine encoding genes induced by these two recombinant proteins after immunization and before infection. Infection of S. veneulensis can make the changes more complicated. Without the direct comparison on the parameters between the immunized without the infection and the immunized with the infection, the results are not convincing although the control groups were included.

3.       The result in Figure 4 wasn’t clearly described. It is not sure in the two control groups, which IgG levels were indicted.

4.       The conclusion wasn’t supported by the results.

5.       The manuscript wasn’t well written, the English need carefully checking and revision.

Author Response

We appreciate the reviewer’s comments. We believe that they have helped us improve our manuscript.

Below, please find our responses to the reviewer’s comments:

  1. In this study, two proteins of S. stercoralis were used, but not against the infection of the same species, other than a different species. However, the information on the sequence identities between Ss-14-3-3 and Sv-14-3-3 as well as between Ss-MA and Sv-MA wasn’t provided. In addition, although the transcription of Ss-14-3-3 and Ss-MA encoding genes is abundant in iL3 of S. stercoralis, it is not known whether this is the same case in S. veneuelensis. Therefore, the reason for selecting these two molecules was not sufficient.

We agree that the best strategy to carry out this kind of vaccination study is to use the same species throughout all the experiments. However, S. stercoralis is mostly restricted to humans, impeding its study in this host for obvious reasons. S. venezuelensis is a parasite of rodent species and, in rats and mice, this parasite exhibit a life cycle similar to S. stercoralis in humans. For this reason, it is often used as a model for human strongyloidiasis, as we did in this study. The aim of our research was to investigate if the 14-3-3 and Major Antigen proteins of S. stercoralis, whose transcripts are very abundant in the transcriptome of the infective L3 stage of this parasite, were able to activate the immune response, using this mouse model. The choice of using the S. stercoralis proteins is justified by the level of conservation among these two species. We agree with the reviewer’s suggestion regarding the inclusion of the data related to the sequence identities. For this, since the sequences of these proteins are not available in the databases for the L3 stage of S. venezuelensis, we have used the nucleotide sequence of the protein fragments to carry out a BLAST against the S. venezuelensis genome to obtain the sequence of the homologous proteins. Then, using the ExPASy bioinformatics tool, the nucleotide sequences were translated to protein sequence. The comparison of the sequence of the proteins from both parasites, revealed 82.4% identity for the MA proteins and 97.5% identity for the 14-3-3 homologues. These data have been now included in lines #136-137.

In addition, we tested the generation of antibodies against S. venezuelensis 14-3-3 and Major Antigen proteins in the mouse model and could show that they increase upon infection (Figure 5). This observation further validates the use of the mouse model to simulate relevant responses comparable to the human infection and suggests that the genes encoding these proteins are likely to be induced upon S. venezuelensis infection. Thus, both the level of similarity between the homologues and the observation that infection induces the production of antibodies specific for these two proteins validate the experimental design chosen here. This comment has been now included in lines #417-422.   

  1. There is a lack of information on the changes of anti-rSs14-3-3 and anti-rSsMA IgG levels and the transcription levels of selected cytokine encoding genes induced by these two recombinant proteins after immunization and before infection. Infection of S. veneulensis can make the changes more complicated. Without the direct comparison on the parameters between the immunized without the infection and the immunized with the infection, the results are not convincing although the control groups were included.

We agree that to include the changes after the immunization and before the infection, would have been appropriate. However, the results show that the infection increases the IgG levels, in the same quantity, for all the groups, immunized or not, demonstrating that the immunization is not inducing a change in the humoral response. It is possible that immunization alone might produce an increase in the IgG levels, but since these S. stercoralis and S. venezuelensis proteins are so similar, we can affirm that the immunization does not potentiate the humoral immune response produced by the infection. In contrast, in the case of some cytokines, a marked difference can be observed between mice only infected and animals previously immunized.

  1. The result in Figure 4 wasn’t clearly described. It is not sure in the two control groups; which IgG levels were indicted.

We agree that the results are not described properly for control groups in figure 5 (previous figure 4).  In the case of control groups, the antibody levels were analyzed by coating the plates with 2 µg/mL of 14-3-3 protein or 2 µg/mL MA protein. The results were very similar for both proteins in the control groups. We have modified new Figure 5 to show both results separately (A and B) one for the ELISA carried out for the 14-3-3 protein and the second one for the MA protein.  This has been specified in the figure legend. The explanation has been included in lines #304-306.

  1. The conclusion wasn’t supported by the results.

We believe that the recombinant Ss14-3-3 and SsMA proteins elicit immunomodulatory effects that might favor the establishment of the female parasites and this contention is supported by the number of females and by the number of eggs per gram, which increased in immunized groups compared to the infection control group. We also believe that these proteins have the potential to be used in ELISA assays to diagnose the disease because the serum of infected mice contain antibodies able to specifically cross-react with Ss14-3-3 and SsMA. Further experiments will confirm this diagnostic capacity. We have removed the last sentence of the conclusions since it was more speculative. Lines #447.

  1. The manuscript wasn’t well written, the English need carefully checking and revision.

The English has been improved in the revised version.

Reviewer 2 Report

vaccines-1825784-peer-review-v1

In this present article, Sánchez-Palencia et al., have studied the immunomodulatory Effect of the Recombinant 14-3-3 and Major Antigen Proteins of Strongyloidesster-starcoralis. This manuscript is lucidly written, and adequate number of experiments were performed. In my opinion, this manuscript can be accepted for publication after addressing the following concerns.

1.     Authors are requested to incorporate a figure corresponding to the purification process of rSs14-3-3 and rSsMA proteins, like PAGE.

2.     What reverse primer is used for qPCR? Section 2.7. Can the author please elaborate on how they estimate cytokines mRNAs. I couldn’t find any gel image.

Author Response

We appreciate the reviewer’s comments. We believe that they have helped us improve our manuscript.

Below, please find our responses to the reviewer’s comments:

  1. Authors are requested to incorporate a figure corresponding to the purification process of rSs14-3-3 and rSsMA proteins, like PAGE.

The purification of rSs14-3-3 and rSsMA in E. coli are shown in a new figure (Figure 1) containing an SDS-PAGE, as requested. Please see lines #228-236.

  1. What reverse primer is used for qPCR? Section 2.7. Can the author please elaborate on how they estimate cytokines mRNAs. I couldn’t find any gel image.

As described in section 2.7., to obtain the total RNA from spleens and to carry out the synthesis of cDNA and the quantitative PCR (TaqMan™ Gene Expression Assays), commercial kits were used and the protocols were applied according to each manufacturer’s indications. The sequences of the primers and the probe used in each qPCR are not available, only the code of the Taqman assay employed for each cytokine. The results are obtained in real time by quantifying fluorescence and the data is expressed as a graph and table of Ct values. It is not necessary to run a gel for the quantification of the cytokine mRNAs using these kits.

Reviewer 3 Report

The aim of this work is to use recombinant proteins from S.stercoralis with the purpose of analyzing their protective capacity against S.venezolensis. For this they use the proteins rSs14.3.3 and rSsAM.

Previously, the authors carried out studies of the epitopes corresponding to these proteins. They found that the mRNAs encoding for both were present at high levels at the infective stage L3. This would suggest, according to the authors, that these proteins could be involved in the skin penetration process that the L3 carry out to start their cycle in humans and that they could participate in the process of modulating the immune response.

Taking into account the clinical relevance of these parasites and the possibility that, especially in people with immunocompromise, they can develop clinical forms of hyperinfection or dissemination, the authors developed an immunization protocol with the recombinant antigens obtained to analyse whether they protected against infection of Sv (Sv in mice develops a cycle similar to that of Ss and is frequently used as an experimental model).

 In the present manuscript they analyse whether they confer protection against infection and/or if they modify the immune response.

The authors have good knowledge of the background on the subject and, as already mentioned, they have experience and publications on the subject.

The methodology used is correct for the objectives of the work. However, it would be worth mentioning why they wait 17 weeks to perform the challenging infection.

The lack of protection of the ags rSs14.3.3 and rSsAM, is verified by the count of eggs in faeces and by the number of parthenogenetic females recovered from the intestine, which is higher in the vaccinated groups than in the only infected ones. Nor do the splenic and intestinal indexs indicate protection.

In relation to the immune response, the animals present high levels of specific IgG related to the infection, but not to the vaccinating antigens.

Regarding the production of cytokines, in the group with rSs14-3-3 mice, an increase in IL-10, TGF-β, IL-13 and TNF-α was recorded. In the group immunized with rSsMA, the authors point out that the increase occurred in IL-10 and IL13. However, IL13 does not differ significantly from controls (the reason for its interpretation should be explained).

In the case of helminth parasites with tissue stages, IL-13 cytokine  favours the restoration of tissue damage, stimulates the differentiation of M1 to M2 macrophages (anti-inflammatories, producers of IL10 and TGF-β. IL13 and IL10 can inhibit the production of proinflammatory cytokines that are normally produced by M1 macrophages.

IL10, in particular, is an enzyme that modulates IR, which might reduce the expulsion of parasites and also restricts the immune response that is harmful to the host. However, no effect in the intestinal infection neither in the intestinal index was registered.

This part of the discussion is correct but too speculative since no experiment is described that would clarify the responsibility of the overexpressed cytokines in the effect of the evolution of the pathology. Experiments with NK IL10 and TGF-Beta KO mice should be included.

Author Response

We appreciate the reviewer’s comments. We believe that they have helped us improve our manuscript.

Below, please find our responses to the reviewer’s comments:

In the present manuscript they analyse whether they confer protection against infection and/or if they modify the immune response. The authors have good knowledge of the background on the subject and, as already mentioned, they have experience and publications on the subject.

  1. The methodology used is correct for the objectives of the work. However, it would be worth mentioning why they wait 17 weeks to perform the challenging infection.

We used a long-term delivery adjuvant system using the immunodulatory AA0029 molecule with antinflammatory properties. Therefore, we increased the vaccination challenge interval to avoid interference with the adjuvant that we observed in a previous study done by our group (Vlaminck J, López-Abán J, Ruano AL, del Olmo E, Muro A. Vaccination against Strongyloides venezuelensis with homologue antigens using new immunomodulators. J Parasitol. 2010;96:643-7). Also, we considered that a wide vaccination-challenge interval is necessary for a vaccine, since in an ideal case, immunity should remain for a long time. In a previous study carried out by our group, we observed better results when the challenge was separated for a long time (Casanueva R, Hillyer GV, Ramajo V, Oleaga A, Espinoza EY, Muro A. Immunoprophylaxis against Fasciola hepatica in rabbits using a recombinant Fh15 fatty acid-binding protein. J Parasitol. 2001;87:697-700). The explanation has been included in lines #161-165.

  1. Regarding the production of cytokines, in the group with rSs14-3-3 mice, an increase in IL-10, TGF-β, IL-13 and TNF-α was recorded. In the group immunized with rSsMA, the authors point out that the increase occurred in IL-10 and IL13. However, IL13 does not differ significantly from controls (the reason for its interpretation should be explained).

It is true that we have pointed out that we observed modest increases in IL-10 and IL13 in the group immunized with rSsMA as compared to controls, but we have mentioned that these results were not statistically significant (lines 322-323). When working with animal models, the variability is, sometimes, too big to make conclusions. That is the reason why we have focused our discussion on the increase in the IL-10, TGF-β, IL-13 and TNF-α that takes place in the group rSs14-3-3 (see lines 378 and 404).

  1. In the case of helminth parasites with tissue stages, IL-13 cytokine favours the restoration of tissue damage, stimulates the differentiation of M1 to M2 macrophages (anti-inflammatories, producers of IL10 and TGF-β. IL13 and IL10 can inhibit the production of proinflammatory cytokines that are normally produced by M1 macrophages. IL10, in particular, is an enzyme that modulates IR, which might reduce the expulsion of parasites and also restricts the immune response that is harmful to the host. However, no effect in the intestinal infection neither in the intestinal index was registered. This part of the discussion is correct but too speculative since no experiment is described that would clarify the responsibility of the overexpressed cytokines in the effect of the evolution of the pathology. Experiments with NK IL10 and TGF-Beta KO mice should be included.

We agree that to prove a causal effect of the overexpression of these cytokines on the evolution of the pathology would require experiments with NK IL10 and TGF-Beta KO mice. We can only report a suggestive correlation. We have modified the text to clarify this point (lines 398-400). We thank the reviewer for this suggestion and, in the future, we will try to understand and directly test the responsibility of the cytokines in the evolution of the pathology of the parasite. It is worth mentioning, however, that under the conditions used in this study, Ss14-3-3 and SsMA do not produce efficient protection against the experimental infection using the S. venezuelensis model. In contrast, the infection was favored by the immunization. Given these results, we were not expecting any variation in the intestinal index because the recombinant proteins did not prevent or increase the local inflammation.

Round 2

Reviewer 1 Report

I am satisfied with the revision and suggest to accept for publication. 

Author Response

Thank you for the time invested and for considering our contributions as positive and accepting the manuscript
